

# Identification of spikes associated with local sources in continuous time series of atmospheric CO, CO₂ and CH₄

Abdelhadi El Yazidi[1]. Michel Ramonet[1]. Philippe Ciais[1]. Gregoire Broquet[1]. Isabelle Pison[1]. Amara
Abbaris[1]. Dominik Brunner[2]. Sebastien Conil[3]. Marc Delmotte[1]. Francois Gheusi[4]. Frederic Guerin[5].
Lynn Hazan[1]. Nesrine Kachroudi[1]. Giorgos Kouvarakis[6]. Nikolaos Mihalopoulos[6]. Leonard Rivier[1].
Dominique Serça[5]

[1]Laboratoire des sciences du climat et de l'environnement, LSCE/IPSL, CEA-CNRS-UVSQ, Université Paris-Saclay,
91191 Gif-sur-Yvette,France

[2]Laboratory for Air Pollution/Environmental Technology, Swiss Federal Laboratories For Materials Science and
Technology, EMPA, Duebendorf, Switzerland

[3]Direction Recherche & Développement, Andra, CMHM, RD960, 55290 Bure France

[4]Laboratoire d'Aérologie, Université de Toulouse, CNRS, UPS, UMR5560, 14 Av. Edouard Belin, 31400 Toulouse,

[5]Geosciences Environnement Toulouse UMR 5563 & UR 234 IRD, Université Paul-Sabatier, Avenue Edouard Belin 14,
F-31400 Toulouse, France

[6]Environmental Chemical Processes Laboratory, Department of Chemistry, University of Crete, 71003, Heraklion,
Greece

*Correspondence to*: A El Yazidi (abdelhadi.el-yazidi@lsce.ipsl.fr)

**Abstract.** This study deals with the problem of identifying atmospheric data that are influenced by local emissions which cause spikes in time series of greenhouse gases and long-lived tracer measurements. We considered three spike detection methods known as coefficient of variation (COV), robust extraction of baseline signal (REBS), and standard deviation of the background (SD), to detect and filter positive spikes in continuous greenhouse gas time series from four monitoring stations representative of the ICOS (Integrated Carbon Observation System) European Infrastructure network. The results of the different methods are compared to each other and against a manual detection performed by station managers. Four stations were selected as test cases to apply the spike detection methods: a continental rural tower of 100 m height in Eastern France (OPE); a high mountain observatory in the south-west of France (PDM); a regional marine background site in Crete (FKL); and a marine clean-air background site in the southern hemisphere in Amsterdam island (AMS). This panel allows to address the spike detection problems in time series with different variability. Two years of continuous measurements of CO₂, CH₄ and CO were analyzed. All the methods were found to be able to detect short-term spikes (lasting from a few seconds to few minutes) in the time series. Analysis of the results of each method leads us to exclude the use of the COV method because of its requirement to arbitrarily specify an a priori percentage of rejected data in the time series, which may over- or under-estimate the actual number of spikes. The two other methods freely determine the number of spikes for a given set of parameters, and the values of these parameters were calibrated to provide a best match with spikes known to reflect local emissions episodes well documented by the station managers. More than 96% of the spikes manually identified by station managers were successfully detected both in the SD and the REBS methods after the best adjustment of parameter values. At PDM, measurements made by two analyzers 200 m from each other allow to confirm that the CH₄ spikes identified in one of the time-series but not in the other correspond to a local source from a sewage treatment facility in one of the



observatory buildings. From this experiment, we found that the REBS method underestimates the number of positive anomalies in the $CH_4$ data caused by local sewage emissions. As a conclusion, we recommend the use of the SD method, which also appears as the easiest one to implement as automatic data processing, for the operational filtering of spikes in greenhouses gases time series at global and regional monitoring stations of networks like ICOS.

## 1 Introduction


Continuous measurements of long-lived greenhouse gases (GHG) such as $CO_2$ and $CH_4$ at ground based monitoring stations are commonly used in atmospheric inversions for the estimation of surface fluxes. The variability of GHG continuous time series reflects atmospheric transport processes and surface fluxes. One difficulty to match these measurements with atmospheric transport model simulations is that they exhibit variability at a wide range of time
scales, which is imperfectly captured by transport models, due to their limited spatial resolution and to uncertain surface emission inventories. In particular, local emissions in the vicinity of stations can have a major influence on concentrations, generating brief but intense positive perturbations, thereafter referred to as "spikes". Every measurement has a specific spatial representativeness, and knowledge of this information, allows a much finer interpretation of the observation. It is desirable to separate in continuous GHG time series the data strongly influenced
by local emissions (fluxes within less than few kilometers) and those influenced by regional (few tens of kilometers) and large scale (hundreds or thousands of kilometers) fluxes and transport. The influence of local fluxes, in particular of nearby point sources of emissions should be filtered out prior to the use of the time series in inversion models if the models do not have the ability to represent it. For instance, a road near a station can emit $CO_2$ causing spikes in the time series, while this road is not accounted for in the emission inventory used in an inversion.

Having empirical information on the representativeness of continuous GHG time series, e.g logbook available for each station, allows for more precise interpretation of the atmospheric measurements, in terms of processes involved in the observed variability. It is interesting for example to assign the contribution of specific sources (e.g. point sources of fossil $CO_2$ emissions, biomass burning events) within the local vicinity of the station. Several methods have been proposed to account for local to regional influences in greenhouse gas observations according to other observables, such
as wind speed and direction (*Perez et al.*, 2012), and tracers like Radon-222 or black carbon (*Biraud et al.*, 2002; *Fang et al.*, 2015; *Williams et al.*, 2016). Air-mass trajectory information are also frequently used (*Ramonet and Monfray*, 1996; *Ferrarese et al.*, 2003; *Maione et al.*, 2008; *Fleming et al.*, 2011; *Perez et al.*, 2012; *Gerbig et al.*, 2006). Other methods based on a statistical treatment of time series (*Giostra et al.*, 2011; *Ruckstuhl et al.*, 2012) are easier to generalize because they require no additional observable. A commonly used strategy by modelers using transport
models of a typical resolution from 10 to 50 km, consists in systematically removing some periods of the day (e.g. night time for surface stations, or day time for mountain sites) in order to filter the influence of non-resolved mesoscale circulations, or vertical transport processes poorly represented by models (e.g. sporadic turbulence in stable or neutral night-time boundary layers).

In this study, our objective is to compare methods that could be used operationally to remove the contaminations from
local sources at continuous measurement stations. Local contamination may be due e.g. to fossil-fuel based power generation at the station facility, local traffic, etc. The short term variations (few seconds to minutes) of greenhouse gases associated to those of local sources have been rarely analyzed, and they have been most of the time averaged with consecutive data. Some studies, though, have been focusing on local emissions on the basis of the detection of short term "spikes" (*Monster et al.*, 2015). "Local" refers here to emissions at less than few kilometers around the station



causing positive short-term spikes of a few seconds to few minutes superimposed on the signal resulting from boundary layer mixing, synoptic transport and regional fluxes. We compare here spike detection algorithms for local sources in greenhouse gases ($CO_2$ and $CH_4$) and long lived tracer time series (CO). The algorithms chosen in this study have been applied to air pollution data (e.g. ultrafine particles, particulate matter, and nitrogen dioxide $NO_2$) having shorter lifetimes than $CO_2$, $CH_4$ and CO (Brantley et al., 2014). In the case of greenhouse gases, spikes can be caused by local

sources but also by the fast transport of remote emissions. Compared to short lifetime species, spikes in GHG are not always larger than the variability associated with synoptic scales. For $CO_2$, uptake by local vegetation may occasionally lead to negative spikes, which will not be evaluated in this study (only positive spikes are considered).

The three spike-detection algorithms: coefficient of variation (COV); robust extraction of baseline signal (REBS); and standard deviation of the background (SD) are described in section 2, then applied to two years of continuous

measurements of $CO_2$, $CH_4$ and CO at four stations representative of the European network of GHG monitoring stations. The results are discussed in section 3. Wherever possible, the ability of an algorithm to successfully detect and remove the effects of local sources and transport is verified using independent information about the presence and position of known local emissions.

**2 Methodology**
We selected four contrasted atmospheric GHG measurement sites operated by LSCE *(Laboratoire des Sciences du Climat et de l'Environnement)*, a tall-tower station in France, a high-mountain station in France, a regional marine background site in Crete, and a marine clean-air site in the southern hemisphere, that provided continuous data from 2013 to December of 2015 (Table 1). Continuous measurements used in this study are averages with one-minute time

resolution, and are processed in near real time by the ICOS Atmospheric Thematic Center (*Hazan et al.*, 2016). The four stations are used in regional and global atmospheric inversions to estimate GHG surface fluxes at a regional and a global scale (*Bergamaschi et al., 2017, Le Quéré et al., 2007, Saunois et al., 2016*).

**2.1 Measurement sites and methods**
**2.1.1 Measurement sites**

**Amsterdam Island** (AMS, 37°48'S; 77°32'E). This marine background station is operated since 1980 to monitor trends of trace gases in the southern-hemisphere mid-latitude clean-air atmosphere**.** The observatory is located on the coast of a small island (55 km²) covered by short grasslands, in the middle of the Indian ocean 3000 km southeast of

Madagascar. Measurements are performed at the *Pointe Bénédicte* site located north of the island, on the edge of a 55-m cliff above sea level. The air is sampled at the top of a 20m high tower. The station contributes to the Global Atmospheric Watch program (WMO/GAW). The data used to feed the WMO/GAW database and estimate the long term trends are filtered according to local wind measurements to avoid the influence of $CO_2$ emissions from the island itself (*Ramonet and Monfray*, 1996).

**Finokalia** (FKL, 35°20'N; 25°40'E). This coastal station is located on the northern coast of Crete, 350 km south of mainland Greece. The nearest city is Heraklion with a population of about 150,000 inhabitants, 50 km west of the station. There is no significant anthropogenic emission within a circle of 15 km around the station (*Kouvarakis et al.*, 2000). The station is on the top of a 230 m hill above sea level, and the air is sampled at the top of a 15 m mast. The dry




season from April to September is associated with strong winds from North and North-west (Central Europe and
Balkans), and the wet season from October to March is associated with air masses from North Africa (South and South-
west winds) in addition to the dominant North-westerly winds. The station is operated by the Environmental Chemical
Processes Laboratory (ECPL) at University of Crete also collects aerosol and reactive gases (*Hildebrandt et al.*, 2010;
*Pikridas et al.*, 2010; *Bossioli et al.*, 2016; *Kopanakis et al.*, 2016).

**Pic du Midi** (PDM, 42°56'N; 0°08'E). This high mountain site is located at 2877m a.s.l on the north and west side of
the Pyrenees range, in southwest France, 150 km east of the Atlantic Ocean and 200 km west of the Mediterranean Sea.
Due to its high elevation, the station often samples tropospheric air from the Atlantic Ocean, but also air masses from
continental Europe in high-pressure conditions over France (north-easterly winds), or from the Iberian Peninsula under
southerly winds. Upslope winds and meso-scale circulations are frequent especially in summer and early autumn,
bringing boundary layer air mostly from southwest France (covered by intensive croplands and forests) (*Gheusi et al.*
*2011; Tsamalis et al. 2014; Fu et al., 2016*).

**Observatoire Pérenne de l'Environnement** (OPE, 48°33'N; 5°30'E). This 120 m tall tower is located in a rural area at
395 m above sea level in the North-East of France (250 km east of Paris). It is located in a transition zone between
oceanic westerly regimes, and easterly winds advecting air from Eastern Europe. The station continuously measures air
quality and greenhouse gases since September 2011 as part of the European infrastructure ICOS. Every hour, ambient
air is sampled for 20 min alternatively at heights of 10, 50 and 120 m on the tower (Table 1).

**2.1.2 Measurement methods**

The gas analyzers used at the four stations are Cavity ring-down spectroscopy instruments (CRDS) (*Okeefe and
Deacon*, 1988), namely Picarro/G2401 analyzers at FKL, OPE and PDM with $CO_2$, $CH_4$ and CO, and Picarro/G2301 at
AMS with $CO_2$ and $CH_4$ (Table 1). The measurement protocols used at the four stations are similar and based on ICOS
recommendations. A calibration using four reference gases is performed every 3 to 4 weeks. Two more reference gases
are analyzed regularly for quality control purposes. The raw data (0.2 to 0.5 Hz) are transferred once per day to a central
server and near-real time (NRT) datasets are available within 24 hours. The NRT data processing (*Hazan et al.*, 2016)
includes automatic filtering of raw data based on the physical parameters of the analyzers (e.g., cavity temperature and
pressure), and threshold values for rejection of outliers This last filter aims to reject aberrant values from the near-real
time dataset. It may happen that it rejects an extreme but real event, for instance due to an urban pollution plume. In
such case the data will be validated afterwards by the station manager. Indeed, after this automatic processing, the
station managers are invited to validate or invalidate data manually using a specific software developed by the ICOS
Atmospheric Thematic Center. For each data manually flagged as invalid, it is required to provide the reason (e.g.
leakage, maintenance, local traffic). This procedure does not ensure the systematic rejection of spikes in the data from
local / regional processes.

Meteorological measurements are also performed at the four stations with barometric pressure, temperature, wind
speed, wind direction and relative humidity. Wind speed and direction are measured using 2D or 3D ultrasonic sensors
installed at the same height of the greenhouse gas measurements. The sensors are adapted to the local weather, for
instance at PDM (2877 m a. s. l) the sensor is heated to avoid icing.




### 2.2 Spike detection algorithms

Three algorithms were tested to detect positive short-duration GHG spikes lasting from a few seconds to a few minutes, using time series of one-minute averaged mole fractions of $CO_2$ (as illustrated in the Supplement, Figure S1), $CH_4$ and CO. The three methods presented in this section, are commonly based on the calculations of the local standard deviations of measurements. A spike is detected when the difference between a determined background and the current data value is above a defined threshold. We will present in this section the corresponding threshold for the three methods.

$CO_2$, $CH_4$ and CO 1-min data were processed using R version 3.1.3 (R Core Team,2015) together with packages openair (*Carslaw and Ropkins, 2015)*, IDPmisc (*Locher et al., 2012*), and ggplot2 (*Wickham et al., 2015*) using the three spike detection algorithms.

### 2.2.1 Coefficient of variation (COV) method

The coefficient of variation (COV) method (*Brantley et al., 2014)* is a modified version of the method presented by Hagler et al. (2010). It was developed to analyze data from a mobile laboratory measuring ultrafine particles concentrations near a road transect (*Brantley et al., 2014)* for peak detection of carbon monoxide which was used as an indicator of the passage of vehicles. In our application we calculate the COV coefficients for $CO_2$, $CH_4$ and CO time series following the next two steps. First, the standard deviation of a moving five minutes' time window (with one

window for each 1-minute data) is calculated (two minutes before and after each 1-minute data point). Second, the standard deviation of each time window is divided by the mean value of the complete time series. The 99$^{th}$ percentile of the COV coefficients is used as a threshold above which a 1-min data is considered to be part of a spike. We also identified as contaminated data all data recorded 2 minutes before and after each contaminated data. The COV method is sensitive to the choice of threshold percentile. In the Supplement we illustrate in Figure S2-A an examples of a spike

detection using the COV method during a CO contamination episode known to be affected by a local fire. One important feature of the COV algorithm, compared to the other methods, is the a-priori definition of the percentage of data to be filtered (threshold percentile), meaning that the number of spike data is not automatically detected.

### 2.2.2 Standard deviation of the background (SD)

The SD method (*Drewnick et al., 2012)* considers that a time series is a combination of a smooth signal superimposed

with a fast variable signal. The variable signal component in our case is related to local emissions causing spikes. To determine the variability of background concentration levels we calculated the standard deviation (σ) of only data falling between the first and the third quartile of all data set. A sensitivity test with various quantile ranges is presented in section 3.1. We then select the first available data, called $C_{unf}$ (un-flagged data, example in the Supplement Figure S2-B) assuming that it is not in a spike. The next data in the time series $C_i$ are evaluated with respect to $C_{unf}$, spikes being

defined by data values higher than a threshold defined as $C_{unf}$ plus an additive value $\alpha * \sigma + \sqrt{n} * \sigma$ : (e.g. the red data point in the Supplement Figure S2-B), where α is a parameter to control the selection threshold, and n is the number of points between $C_{unf}$ and $C_i$. The value of α depends on the time-series variability. A sensitivity analysis to the influence of α is presented in section 3.1. We set a default value of α=1 for $CO_2$ and $CH_4$, and α=3 for CO (*Drewnick et al., 2012)*. The lower value for $CO_2$, and $CH_4$ is justified in section 3.1. The integer n brings a temporal information





about the evolution of the time-series. Indeed, while identifying a spike $C_i$, the next data is evaluated against $C_{unf}$ using an increased threshold to take in consideration the variability of the baseline during the spike event. If $C_i$ is lower than the threshold from equation (1), it is considered as non-spike, and becomes the new reference value $C_{unf}$. The following data will then be compared to this updated $C_{unf}$.

$$C_i \geq C_{unf} + \alpha * \sigma + \sqrt{n} * \sigma \qquad (1)$$

The SD method was applied over one-week time windows, i.e. the standard deviation $\sigma$ over a week is used for threshold calculation. Using a longer period (e.g. one year) would give more weight on the seasonal and long-term variabilities which are not relevant to identify short term spikes using the one-year standard deviation.

### 2.2.3 Robust extraction of baseline signal (REBS)

The REBS method (*Ruckstuhl et al. 2012*) is a statistical method based on the calculation of a local linear regression of the time-series over a moving time-window (characterized by a duration called the "bandwidth"), to account for the slow variability of the baseline signal, while outliers lying too far from the modelled baseline are iteratively discarded. The bandwidth *h* must be wide enough to allow for a sufficiently low fraction of outliers within h. The REBS code used here is based on the *rfbaseline* application developed in the IDPmisc package (*Locher, et al., 2012*) in R software. It is a
modified version of the robust baseline estimation method developed to delete baseline from chemical analytical spectra (*Ruckstuhl et al., 2001*). The REBS method was applied at the high-alpine Jungfraujoch site (Switzerland, 3580 m a.s.l.) and proved robustness to estimate the background measurements of greenhouse gases (*Ruckstuhl et al. 2012*). The REBS method considers that greenhouse gases time-series are composed of a background signal, plus a regional contribution which may also include local effects (spikes) and measurement errors. The main difficulty is to correctly
define the baseline signal of the measured time-series. To achieve this goal, the choice of the bandwidth value is important. In the Jungfraujoch study, the baseline signal was defined as the smooth curve retrieved from REBS technique (*Ruckstuhl et al. 2012*) using a bandwidth of 90 days, in order to distinguish the contribution of regional emissions that add to the slow seasonal variability. Since, in our study, the targeted spikes last few seconds to few minutes, we chose to calculate the baseline using a bandwidth of 60 min to detect spikes of a few minutes (maximum 5
minutes). The threshold for spike detection in REBS is based on the calculation of a scale parameter $\beta$ which represents the standard deviation of data below the baseline curve, called $\hat{g}(t_i)$. All measurements $Y(t_i)$ that satisfy $Y(t_i) > \hat{g}(t_i) + \beta * \gamma$ are classified as locally contaminated (illustration in the Supplement Figure S2-C). $\beta$ is a parameter to adjust the filtering strength. Ruckstuhl et al. (2012) set $\beta$ =3 for the detection of polluted data. For our purpose, a sensitivity test with different values of $\beta$ is carried out (section 3.1).






## 3 Results

### 3.1 Optimization of the SD and REBS methods

#### 3.1.1 Sensitivity to the parameters of the SD method

We conducted sensitivity tests in order to evaluate the influence of the two parameters $\alpha$ and $\sigma$ used in the SD method. For $\alpha$ we tested values ranging from 1 to 3. Here, we present only the results for $\alpha=1$ and $\alpha=3$. For $\sigma$ we compared the results calculated with $\sigma$ based on 50% of one-week data, data between the first and third quartile (scenario $\sigma_b$), and for all the data of the week (scenario $\sigma_t$). We studied four configurations (two values of $\alpha$ with $\sigma_b$ or $\sigma_t$) on one-minute data every week at the four stations. Figure 1 shows an example of spikes detected by SD at FKL on December 16, 2014,

corresponding to a known waste-burning episode reported by the station manager. The station logbook mentions waste burning occurring nearby the station between 6:30 am and 8:30 am, shown by a purple bar in Figure 1. The blue area in Figure 1 shows the CO data between first and third quartile leading to a $\sigma_b = 3.6$ nmol.mol$^{-1}$. Considering all the data, we obtain a three time higher standard deviation: $\sigma_t = 12.5$ nmol.mol$^{-1}$. The SD method with $\alpha=3$ and $\sigma_b=3.6$ nmol.mol$^{-1}$ selects two 1-min data as spike as illustrated by the orange dots falling within the observed fire episode in Figure 1.

With $\alpha=3$ and $\sigma_t=12.5$ nmol.mol$^{-1}$, the method fails to detect any spike, indicating that the threshold value was too high. With $\alpha=1$ and $\sigma_b$ the SD method selects 44 additional 1-min spikes compared to $\alpha=3$ (data not reported as contaminated by the station manager). In both cases ($\alpha=1$ or $\alpha=3$) and $\sigma_t$ lead to very high threshold, and an underestimation of the number of spikes detection, since $\sigma_t$ includes the spike variabilities. Based on this sensitivity test against a known local emission episode, we definitively rejected the use of $\sigma_t$ scenario.

Table 2 represents the percentage of contaminated data detected over one year at the four sites, in the four tested configurations. As can be seen, using all 1-min data to calculate $\sigma_t$ leads to a higher threshold and consequently to less data detected as contaminated, spikes in other words. On average over the four stations and the three species, switching from $\sigma_b$ to $\sigma_t$ decreases the percentage of spikes by a factor 15 $\pm16$ (Table 2). Setting $\alpha=3$ increases the threshold and also decreases the number of spikes by on average a factor of 5 $\pm7$ (Table 2). The parameter $\alpha$ is related to the

variability of the time-series. Since our study aims to provide recommendations for automatic data processing of a monitoring network like ICOS in Europe, we want to keep the same set of parameters for all the stations of the network for each species. However, all the tests conducted in the present study have shown that it was not optimal to use the same parameter for CO time series compared to $CO_2$ and $CH_4$ ones. Setting a lower $\alpha$ for CO lead to the over-estimation of the number of spikes in the time series. This must result from the different variabilities of those trace gases. For

instance, the ratio between hourly and minute scale variabilities (characterized by standard deviations) for the sites used in this study, is on average two times smaller for CO compared to $CO_2$ and $CH_4$. As recommended in Brantley et al. (2014) and Drewnick et al. (2012), we decided to keep $\alpha = 3$ for CO, and set $\alpha = 1$ for $CH_4$ and $CO_2$ because of their lower variability.

#### 3.1.2 Sensitivity to the parameters of the REBS method

In order to evaluate the sensitivity of spikes to the parameter $\beta$, we tested values of $\beta$ ranging from 1 to 10. In this sutdy, we present the REBS method using the default value $\beta=3$ as proposed by (Ruckstuhl et al., 2012) in Junfraujoch, compared with the optimal value for our purpose $\beta=8$. The resulting spike selection at FKL (during a





local fire episode) is shown in Figure 2. By setting $\beta$ =3, the REBS method detects the spike during the episode but

it also finds other events which do not appear to be associated with evident contaminations (Figure 2). With $\beta$ =8, the REBS correctly detects spikes during the fire episode (orange points in Figure 2). We further compared these two values of $\beta$ at the four stations every week, and report spikes detection statistics in Table 3. About 10 times more spikes for CO, and 5 to 7 times more for $CH_4$ and $CO_2$ were detected by the REBS method with $\beta$ =3 compared to $\beta$ =8. Using $\beta$ =3, we detected more than 2% of spikes for all species and up to 7% for $CO_2$ at AMS. Using

$\beta$ =8 these percentages are reduced to 0.1% and 1%, respectively (Table 4), in better agreement with spikes manually reported by site managers.

Based on these sensitivity tests for the SD and REBS parameters, and the a prior estimation of the percentages of spikes manually detected by site managers, we apply the SD method with $\sigma_b$ and $\alpha$=3 for CO, and with $\sigma_b$ and $\alpha$=1 for $CO_2$ and $CH_4$. For the REBS method we use $\beta$ =8.


### 3.2 Statistics of the three spike detection methods

The statistics for local spikes detection with the three methods are given in Table 4. With COV we detect an average of about 2% of spikes with the 99th percentile threshold for all stations and species (section 2.2). With the methods SD and REBS more variable percentages of spikes are found depending on the trace gas variabilities at each station. The

percentages of contaminated data range from 0.1% for $CO_2$ at AMS, to 7% for $CH_4$ at PDM. The value of 7% detected for $CH_4$ at PDM is higher than at all other sites / species, and reveals the influence of a source of methane on the site (see below and next paragraph). For OPE, we found a significant percentage of spikes (between 1 and 2%) for all species, which may be explained by the higher number of local emission sources compared to other stations located in more pristine environments. At FKL and AMS we obtain different percentages of spikes between SD and REBS for

$CO_2$. In fact, we assume that this difference can be related to the sea land circulation, when winds turn, leading to a fast change in atmospheric concentrations. For FKL, AMS, and PDM, the percentage of spikes found with the SD and REBS methods vary around 1% with the exception of $CH_4$ at PDM where both SD and REBS detect high percentages of spikes (7% for SD method and 2.3% for REBS method). This is not expected for a high mountain station. The results of the field campaign organized at PDM in 2015 (section 3.3) revealed the influence of a local water treatment facility

in a building of the station, producing $CH_4$ (see section 3.3).

Generally, the methods SD and REBS detect automatically spikes. However, the COV method requires a prior knowledge of data sets and the approximate number of data to be filtered. Because of this limitation for automatic spike detection we have discarded the COV method from further tests for the selection of the most reliable method for spike detection.


### 3.3 Comparison of SD and REBS methods to detect $CH_4$ spikes at the PDM clean-air mountain station

In this section we use field campaign data involving two instruments at PDM to study the efficiency of the SD and REBS methods. As noted above, the PDM $CH_4$ record shows many spikes (duration of a few minutes) superimposed on

low frequency variations in the background signal (time scales from hours to days). Such spikes were not observed in the $CO_2$ monitoring with the same analyzer. The SD method detects 20 times more spikes for $CH_4$ than for $CO_2$ at PDM



ICOS site (Table 5). Looking for all possible local methane emissions at the site, we identified a small sewage treatment facility located about 20 m below the air intake of the analyzer (called AN-1) to be responsible for local $CH_4$ production. A test campaign was then organized between July and August 2015 with a second analyzer (called AN-2)

installed in another building at the opposite side of the station platform, 200 m away from the location of AN-1 (a picture of the location of the two buildings is presented in the Supplement Figure S3). The two analyzers were installed to measure simultaneously $CH_4$ and $CO_2$ molar fractions from first of July to $31^{th}$ of August. The $CH_4$ and $CO_2$ time series from analyzers AN-1 and AN-2 running in parallel are presented in Figures 3 and 4.

We applied the SD and REBS methods to the $CH_4$ and $CO_2$ time-series from both analyzers. For $CH_4$, analyzer AN-2

located away from the sewage shows much less spikes than AN-1. For instance, between early July and late August 2015, there is more than 12% of contaminated data with the SD method, and 3% with the REBS method in the AN-1 record, against only 0.8% with SD and 0.7% with REBS with the AN-2 instrument (table 5). Considering that the two analyzers are measuring ambient air sampled 200 m apart, this large difference is clearly due to the local emission from the sewage facility. Interestingly, for $CO_2$ we detect more spikes in AN-2 than in AN-1 (Figure 4). More than 1% of $CO_2$

spikes were found in the AN-2 record compared to 0.5% for AN-1 (Table 5, Figure 4). This is explained by the proximity of a diesel generator to AN-2, although this generator is used only a few hours every month (especially in case of electrical storm). Both SD and REBS detect the same $CO_2$ spikes in both AN-1 and AN-2 time-series (Figure 4). Running two analyzers in parallel allowed us to understand the unexpected high spikes percentage of the $CH_4$ time-series at Pic Du Midi. Both SD and REBS confirm the frequent contamination of the $CH_4$ time series of AN-1 since

2014, and show a good ability to detect the spikes, yet with significant differences regarding the percentage of data detected as contaminated. Considering that AN-2 analyzer provides less contaminated $CH_4$ time series, we have used this experiment to compare between the two methods and select which one performs better for $CH_4$ spikes at PDM. Figure 5 and 6 represent the $CH_4$ and $CO_2$ measurements of AN-1 and AN-2. Black data points are the sampled data; and the green ones are the filtered data using the SD (A) and REBS (A') methods. For AN-2, $CH_4$ concentrations (black

data point in Figure 5) rarely exceed 1950 $nmol.mol^{-1}$, whereas for AN-1, it exceeds 2000 $nmol.mol^{-1}$ (black data point), and occasionally reached almost 2200 $nmol.mol^{-1}$ (unexpected high value for a clean-air mountain station). SD and REBS methods both detect all contaminated data that range between 1980 and 2200 $nmol.mol^{-1}$ for AN-1. The differences between the two automatic methods are more important for data that are below 1980 $nmol.mol^{-1}$. In fact, the filtered data (green data point) using the SD method fits better the 1:1 correlation line with the less contaminated

analyzer than the REBS method (Figure 5). The REBS method underestimates the lower part (foot) of the spikes (contaminated data that range between 1900 and 1980 $nmol.mol^{-1}$, Figure 3-A' AN-1). On the other hand, for $CO_2$ the two methods detect nearly the same spikes, as shown in Figure 4, and provide similar filtered time series (green data point in Figure 6). How can we explain the insufficient performance of the REBS method to detect the lower part of the $CH_4$ spikes? This method defines spikes using the estimated baseline (*Ruckstuhl et al. 2012*). When the population of

contaminated data is high, the baseline is flawed due to the influence of spikes, and the baseline determination will be overestimated. In Figure 5, we can clearly notice the missed detection of many contaminated data by REBS method, due to the high values of the baseline. The SD method, despite its simplicity thus appears to detect correctly most of the local spikes at PDM, even if a slight underestimation of contaminated $CH_4$ data remains even after data filtering (deviation from the 1:1 line). This underestimation is related to the spikes residues (spikes foot that persist after

filtering).





### 3.4 Comparison between automatic and manual spike detection

In this section we analyze how the SD and REBS methods detect spikes of $CO_2$, $CH_4$ and CO, that were independently identified by the station staff and related to a known local source of contamination at FKL and PDM.

At FKL the contamination events reported by the site manager are associated with local fires nearby the station. The technical staff recorded dates of burning (plant residues from nearby grazing land) which could lead to significant emissions of trace gases, especially CO and $CO_2$. It should be noted, however, that this information is not exhaustive in a sense that the person in charge does not necessarily have information on all burning events. We have matched the trace gas time series with the logbook information showing 17 days with local burning events between 2014 and 2015.

We applied the SD and the REBS methods over one-week time windows containing each burning event. First, we run the algorithms separately on the three species ($CH_4$, $CO_2$, and CO). Then, if the algorithm detects a spike in at least one species, we consider as spikes data for all other species as well. In the case of spike detection related to waste burning events we can use the CO measurements as a reference. Several studies demonstrate that fires plumes lead to strong enhancements of CO concentrations in the atmosphere (*Forster et al., 2001*). As an example, the CO spikes during local

fire episodes can exceed 100 $nmol.mol^{-1}$ in less than one minute at FKL. In the Supplement, Figure S4 shows an example of the SD method applied on a fire episode between 03:00 pm and 04:00 on November $6^{th}$, 2014. The spike occurred simultaneously for the three species CO, $CO_2$ and $CH_4$, with a similar pattern. The same spike was identified by the station manager, demonstrating the ability of an automatic method to detect a real local contamination event.

The SD method and REBS method were able to detect the 17 events associated with local fires. Figure 7-A represents

the number of contaminated data (minute averages) detected by the automatic methods (SD and REBS) and manual flagging by the station staff. The numbers of selected data are split into three concentration ranges. The two automatic methods and the manual flagging detect the same number of contaminated data for CO classes higher than 400 $nmol.mol^{-1}$. We have an excellent agreement for the spikes with the highest concentrations. For the low concentration spikes (< 400 $nmol.mol^{-1}$), the automatic methods are less selective than the manual flagging. In Figure 8 we show

another example of contaminated data detected by manual flagging at FKL, compared to spikes retrieved by the SD and REBS methods. When the difference between uncontaminated (identified as reference) and spike data is not significant compared to a certain standard deviation threshold, the methods may thus fail. The data highlighted by the blue circle in Figure 8 give an example of spikes for automatic methods diverge from the manual information. These data are either close to the baseline REBS selection (Figure 8 C), or close to the $C_{unf}$ value for the SD method (Figure 8 B). Those are

the cases where the automatic methods may underestimate the contaminated data, especially spike foot. At this point it is important to note that the person in charge of data flagging selects spikes using a known period (from a starting to an ending time).

A second comparison study between automatic methods and manual detection has been performed at PDM using CO time series from December to February 2014. During winter, the station experienced several snowfall episodes and

snow was removed with a diesel powered snow blower. This operation influenced the GHG concentrations and leads to sharp spikes easily observed in the CO time series (Supplement Figure S5). The site managers eliminated manually all these data. For comparison, (as illustrated in the Supplement Figure S5) we display the spikes detected from December to February 2014 by the SD and the REBS methods. Most of the spikes are successfully detected by the SD and the REBS methods. Figure 7-B represents the number of contaminated data detected by SD in red and REBS in green, and

manual flagging in blue. Similar to the FKL local fires, the SD and the REBS methods detected the same number of spikes than the manual selection for high concentrations. 857 contaminated data are detected by the SD method (same





as the PI) for concentration higher than 400 nmol.mol-1, and 828 data are detected by the REBS method. The main difference between the automatic methods and the manual flagging are related to the lower part of the spikes. For 2861 data (CO < 400 nmol.mol$^{-1}$) flagged manually by the PI station, the SD method detects 2270 data, when the REBS

method detects only 1799 data. In fact, for moderate spikes the SD method selects 70% of contaminated data according to the PI, when the REBS method retrieves only 60%.

**3.5 Influence of the spike detection on hourly averages:**

In this section we estimate the impact of the spike detection on data used for atmospheric inversions, which are typically hourly or half-hourly averages. For this purpose we have calculated the differences between the hourly averages of the filtered and non-filtered time-series. In table 6, we present the number of hours in which we filtered at least one-minute data, and for each species, we classified the results into three intervals. For $CO_2$, the first interval represent the values lower than 0.5 µmol.mol$^{-1}$, the second interval is for differences between 0.5 and 1 µmol.mol$^{-1}$, and

the third one stands for the higher differences (values more than 1 µmol.mol$^{-1}$). For $CH_4$ and CO we set the first interval is for values lower than 5 nmol.mol$^{-1}$, the second interval represents the data between 5 and 10 nmol.mol$^{-1}$, and the third one is for differences higher than 10 nmol.mol$^{-1}$. Most of the differences between filtered and non-filtered hourly data vary between 0 and 0.5 µmol.mol$^{-1}$ for $CO_2$, and between 0 and 5 nmol.mol$^{-1}$ for $CH_4$ and CO. For $CO_2$ in AMS station, the SD method detects 1454 one minute data (table 4), which occur in 104 hours during the three years of

measurements. 62% of those hours are characterized by a difference up to 0.5 µmol.mol$^{-1}$, and 18% show more than 1 µmol.mol$^{-1}$ of difference. For $CH_4$ measurements in AMS, the 8801 contaminated data detected by the SD method (table 4) occur in only 21 hours, and this modifies the hourly averages by 5 nmol.mol$^{-1}$ as a maximum. For the four sites, we notice similar effect on the hourly averages. Most of the impacted hours are characterized by a difference within the first interval (0.5 µmol.mol$^{-1}$ for $CO_2$; 5 nmol.mol$^{-1}$ for $CH_4$ and CO). However, for OPE we observe higher

differences with 53%, 36% and 47% of the impacted hours in the highest interval respectively for $CO_2$, $CH_4$ and CO. This feature is probably related to the higher number of the nearby local emission sources nearby OPE site compared to the other stations located in more pristine environments. The Figure S6 shows a decrease of the number of impacted hours for higher intervals (the same pattern as the three other stations).Overall, the aggregation of the filtered measurement at the hourly time scale showed a relatively weak impact of the filtered data for background sites, but a

more significant effect for stations located closer to local sources.





## 4. Conclusion

The development of regional networks for monitoring greenhouse gases (GHG) and related tracer concentrations leads to an increasing number of continuous measurement stations, especially in continental areas. For example, the European ICOS research infrastructure is developing a network of tall towers for very precise GHG measurements across the European continent. It is thus important to characterize the representativeness of each individual measurement, in order to separate spikes from local emissions that should not be used in studies aiming at constraining regional fluxes.

We addressed the problem of identifying concentration spikes of few minutes duration in GHG continuous series by applying automatic detection methods (COV, SD and REBS) previously used for atmospheric pollution but not systematically for GHG time series. Stations with different regimes of variability where local emission sources are identified without ambiguity (engines / waste near the station buildings, or fires nearby) are chosen to evaluate the performance of the automatic methods against spikes manually identified by station managers. The COV algorithm can be considered as a semi-automatic method since it requires an a-priori choice of a percentage of data rejected as spikes. We tested the COV method with a percentage of 1% of spike data for all species and for all stations. This limitation made the COV method less flexible and informative for universal automatic spike detection across different sites. For the two fully-automatic methods (SD and REBS) we performed several sensitivity tests in order to recommend the best set of parameters for our 4 chosen stations considered to be representative of most ICOS stations (disregarding those located in sub-urban environments).

The application of the automatic methods on contaminated time-series at the Pic Du Midi observatory showed the ability of SD and REBS to detect real spikes on the $CH_4$ time series caused by the sewage treatment of the observatory. Nevertheless, significant differences regarding the rejection percentage were noticed between the methods. Both methods have a tendency to unduly keep a certain fraction of the spike base (lowest concentrations in spikes). REBS is worse than SD in this respect. In the REBS method, when the percentage of spikes is high, the baseline determination is biased toward high concentrations, leading to underestimate spike anomalies above this baseline. However, the SD method correctly detects most of the contaminated data. The comparison between SD and REBS and the manual flagging showed a good agreement with an overall percentage of 70% of successful spike data detection for SD, and 60% for REBS, at two stations (FKL and PDM) where local contaminations are well identified by the local staff. These two automatic algorithms detect short-term spikes, allowing for a more consistent and automatic filtering of the time series even if they select less spikes data the manual flagging. The estimation of the impact of the spike detection on data used for atmospheric inversions showed a relatively weak impact of the filtered data for background sites, and a more significant effect for stations located closer to local sources. The SD method is found to be efficient and reliable for the purpose of spike detection. It has been proposed for operational implementation in the ICOS Atmospheric Thematic Center Quality Control (ATC-QC) software to perform daily spike detection of the near-real time dataset of continuous ICOS stations. The first step will be to run the SD method in a test mode over all ICOS stations and compare with manual detection when available, in order to set optimal values of parameters. This analysis can be complemented with wind speed and direction data in order to possibly attribute spikes to fixed local sources.





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




| Site | Measured spices | Instrument | Longitude | Latitude | Ground level (m asl) | Sampling hight (m agl) | Starting date | End date |
|---|---|---|---|---|---|---|---|---|
| Pic du Midi (PDM) | CO, $CO_2$, and $CH_4$ | Picarro / G2401 | 0°08'E | 42°56'N | 2877 | 10 | 2014-05-07 | 2015-12-31 |
| Observatoire Pérenne de l'Environnement (OPE) | CO, $CO_2$, and $CH_4$ | | 5°5'E | 48°55'N | 395 | 10, 50 and 120 | 2013-03-07 | |
| Finokalia (FKL) | CO, $CO_2$, and $CH_4$ | | 35°20' E | 25°40' N | 230 | 15 | 2014-06-05 | |
| Amsterdam (AMS) | $CO_2$, and $CH_4$ | Picarro / G2301 | 37°48' E | 77°32' S | 55 | 20 | 2013-01-01 | |

**Table 1: Measurement sites characteristics**






| Site | Spices | Contaminated data percentages (%) | | | |
|---|---|---|---|---|---|
| | | $\sigma_b$ scenario | | $\sigma_t$ scenario | |
| | | $\alpha = 1$ | $\alpha = 3$ | $\alpha = 1$ | $\alpha = 3$ |
| AMS | $CH_4$ | 0.03 | 0.01 | 0.006 | 0.003 |
| | $CO_2$ | 0.07 | 0.03 | 0.01 | 0.006 |
| FKL | $CH_4$ | 0.2 | 0.02 | 0.02 | 0.002 |
| | $CO_2$ | 0.1 | 0.04 | 0.01 | 0.002 |
| | CO | 3 | 0.4 | 0.3 | 0.07 |
| OPE | $CH_4$ | 0.7 | 0.3 | 0.06 | 0.01 |
| | $CO_2$ | 0.8 | 0.04 | 0.02 | 0.01 |
| | CO | 0.9 | 0.4 | 0.1 | 0.02 |
| PDM | $CH_4$ | 6 | 2 | 1 | 0.1 |
| | $CO_2$ | 0.2 | 0.05 | 0.02 | 0.005 |
| | CO | 3 | 0.1 | 0.04 | 0.004 |

**Table 2: Sensitivity of SD method spike detection for two sets of α (α=1 and α=3), and for two range of background data interval ($\sigma_b$ and $\sigma_t$ scenario) for the four stations and all species.**





| Sites | Species | Contaminated data percentages (%) | |
|-------|---------|:---:|:---:|
| | | $\beta$ =3 | $\beta$ =8 |
| AMS | $CH_4$ | 2 | 0.3 |
| | $CO_2$ | 7 | 1 |
| FKL | $CH_4$ | 5 | 0.8 |
| | $CO_2$ | 4 | 0.6 |
| | CO | 1 | 0.1 |
| OPE | $CH_4$ | 2 | 0.4 |
| | $CO_2$ | 2 | 0.4 |
| | CO | 1 | 0.1 |
| PDM | $CH_4$ | 8 | 2 |
| | $CO_2$ | 5 | 0.7 |
| | CO | 2 | 0.2 |

**Table 3: Sensitivity of REBS spike detection method for two sets of $\beta$ ( $\beta$ =3 and $\beta$ =8) for the four stations and all**

**species.**





| Sites | species | SD | | REBS | | COV | |
|---|---|---|---|---|---|---|---|
| | | Percentage (%) | Number of detected data | Percentage (%) | Number of detected data | Percentage (%) | Number of detected data |
| AMS | $CH_4$ | 0.6 | 8801 | 0.2 | 3318 | 2.1 | 29315 |
| | $CO_2$ | 0.1 | 1454 | 1.7 | 24210 | 1.8 | 24672 |
| FKL | $CH_4$ | 0.3 | 2096 | 1 | 7680 | 2 | 14657 |
| | $CO_2$ | 0.1 | 1052 | 0.6 | 4831 | 1.9 | 14295 |
| | CO | 0.2 | 1618 | 0.1 | 1002 | 2.1 | 15617 |
| OPE | $CH_4$ | 1.8 | 5473 | 1 | 2987 | 1.3 | 3864 |
| | $CO_2$ | 1.1 | 3296 | 1 | 2749 | 1.5 | 4186 |
| | CO | 1.3 | 3777 | 1.1 | 3120 | 1.4 | 4118 |
| PDM | $CH_4$ | 7 | 56548 | 2.3 | 19056 | 1.8 | 14243 |
| | $CO_2$ | 0.3 | 2567 | 1 | 8757 | 1.9 | 15618 |
| | CO | 0.2 | 1970 | 0.2 | 1348 | 2 | 16603 |

**Table 4: percentage (rounded to one decimal) and number of contaminated data detected by SD, REBS, and COV method overall stations (AMS, FKL, OPE and PDM) and for the three species CO, $CO_2$ and $CH_4$.**




| | | ICOS site | | TDF site | |
|---|---|---|---|---|---|
| | | SD | REBS | SD | REBS |
| $CH_4$ | Percentage (%) | 13 | 3 | 0.8 | 0.7 |
| | Number of contaminated data | 10244 | 2396 | 684 | 602 |
| $CO_2$ | Percentage (%) | 0.2 | 0.5 | 1.1 | 1.4 |
| | Number of contaminated data | 158 | 390 | 849 | 1050 |

**Table 5: percentages and number of contaminated data detected by SD, REBS methods for $CO_2$ and $CH_4$ at PDM.**





| | | $CO_2$ ( mol.mol$^{-1}$) | | | $CH_4$ (nmol.mol$^{-1}$) | | | CO (nmol.mol$^{-1}$) | |
|---|---|---|---|---|---|---|---|---|---|
| | ]0-0.5[ | [0.5-1[ | >=1 | ]0-5[ | [5-10[ | >=10 | ]0-5[ | [5-10[ | >=10 |
| AMS | 64 (62 %) | 21 (20%) | 19 (18%) | 21 (100%) | 0 | 0 | | | |
| FKL | 133 (89%) | 12 (8%) | 5 (3%) | 134 (88%) | 11 (7%) | 7 (5%) | 218 (93%) | 8 (3.5%) | 8 (3.5%) |
| PDM | 522 (92%) | 30 (5%) | 16 (3%) | 4696 (78%) | 741 (12%) | 623 (10%) | 518 (99%) | 4 (0.8%) | 1 (0.1%) |
| OPE | 36 (28%) | 24 (19%) | 69 (53%) | 53 (54%) | 10 (10%) | 36 (36%) | 107 (45%) | 20 (8%) | 111 (47%) |

**Table 6: Classification of the number of hours in which the SD method filtered at least one-minute data point for CO, CO2, and CH4 at the four sites. The intervals represent the differences between filtered and the non-filtered time-series averaged at a hourly scale in (µmol.mol$^{-1}$) for CO2 and (nmol.mol$^{-1}$) for CO, and CH4.**






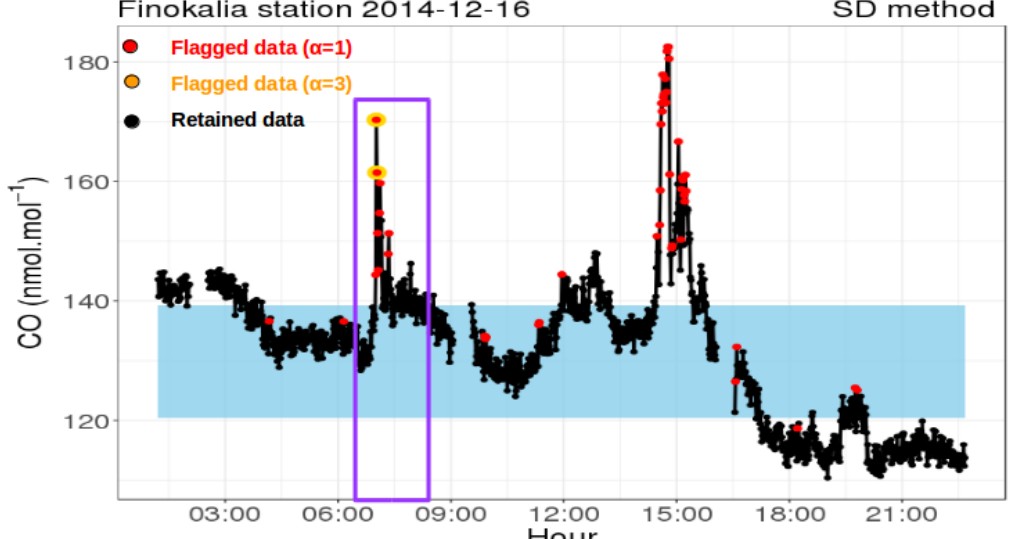

**Figure 1: comparison between two sets of α parameter for SD method. Red color represents detected spikes for α=1, orange data are the detected spikes for α=3. The blue area shows the data between the first and the third quartile (q1=0.25, and q2=0.75).**






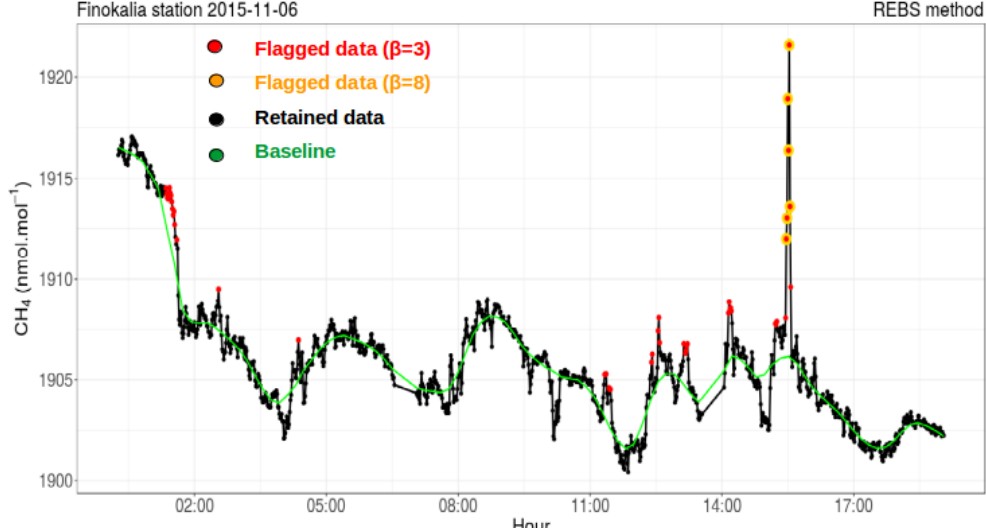

**Figure 2: comparison between two sets of ß parameter for REBS method. Red represents detected data for $\beta$ =3, orange are the detected data for $\beta$ =8, applied on FKL measurement 6th of November 2014.**





**Figure 3: AN-1 CH$_4$ measurement at T55 building for A and A', and AN-2 TDF building for B and B'. Black data points are the retained measurements, red points represent the flagged using SD method for A and B, and REBS method for A' and B'**






**Figure 4: AN-1 CO$_2$ measurement at T55 building for A and A', and AN-2 TDF building for B and B'. Black data points are the retained measurements, red points represent the flagged using SD method for A and B, and REBS method for A' and B'**



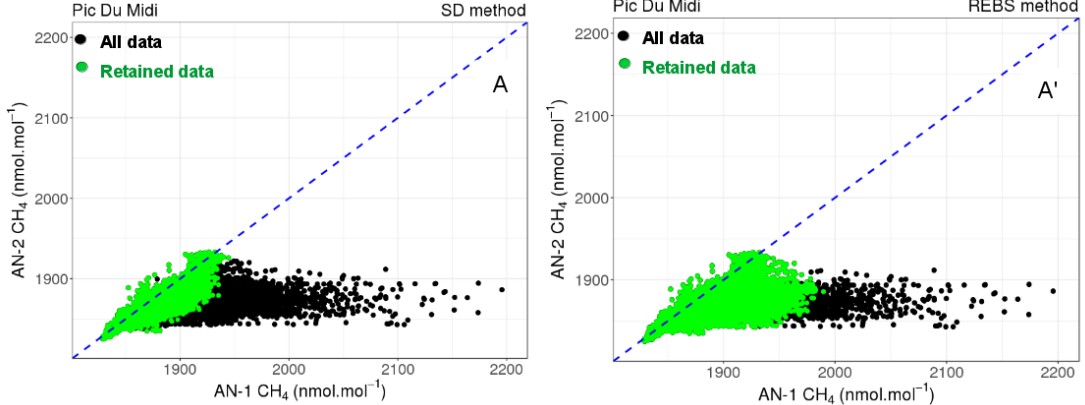

**Figure 5: plots of CH$_4$ measurements of AN-1 against AN-2. All data are in black, and the green points represent the retained data using SD method for A and REBS method for A'**

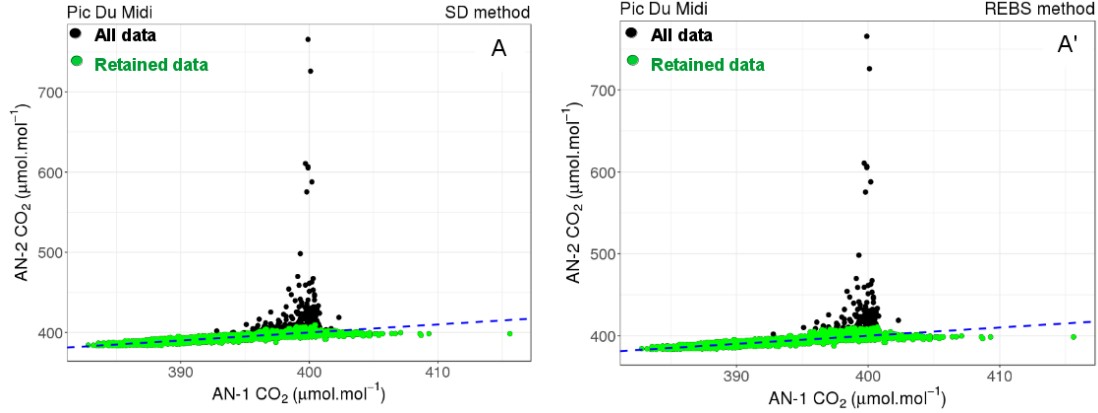

**Figure 6: plots of CO$_2$ measurements of AN-1 against AN-2. All data are in black, and the green points represent the retained data using SD method for A and REBS method for A'**





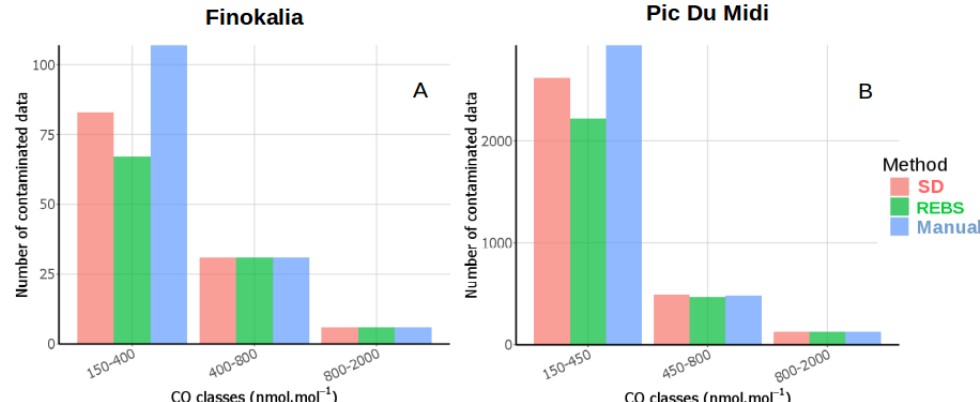

**Figure 7: Number of flagged CO measurements using manual method (blue), SD method (red), and REBS method (green) for Finokalia (A) and Pic Du Midi (B) .**






**Figure 8: Example of a spike detection using manual (A), SD (B), and REBS (C) methods during a known biomass burning event at Finokalia.**