# Peer review of "Identification of spikes associated with local sources in continuous time series of atmospheric CO, CO2 and CH4"

_Atmospheric Measurement Techniques, 2017_

## Referee Comment (RC1) · Anonymous Referee #2 · 18 Oct 2017

The manuscript has been improved since the first submission in acp. The style is more accurate and more information has been added.

However in my opinion there are still things that need to be clarified:

In section 3.1.2 the authors indicates that test values ranging Beta from 1 to 10 had been done. The results of these tests should be presented (at least in the supplementary information). Otherwise, the affirmation that the optimal value is Beta=8 seems arbitrary.

In table 4 it could be interesting to show also the number of spikes manually reported by site managers, at least for the sites where this information is available. (in AMS are

close to 0.1% for all species and 1% for CO2 according to text).

The overestimation is not reflected in the paper (not even in section 3.3). Probably is very low for both SD and REB methods (as in fact they underestimate the number of events) but needs to be quantified. For example, in section 3.4: was there any event not considered by manual spike detection that was considered as spike by any of the 2 detection methods?

In Table 6, it will be interesting to add the percentage for each classification on the whole series (e.g. OPE filtered data is, in 53% of cases, higher than 1ppm from non filtered. But which percentage of hours this represents on the whole series?)

I figure S6: please, use percentage of hours instead of number of hours. Graph will also be clearer with another X scale. Moreover, as in the text is said the figure is similar for other stations, it will be interesting to add the histograms of all sites.

The reference Foster et al. does not seems to be related with this article.

---

## Referee Comment (RC2) · Anonymous Referee #1 · 9 Dec 2017

**Introduction**

In this paper an attempt is made to systematically separate spikes in records of ambient greenhouse gas mole fraction observations caused from local emissions that should not be used in studies aiming at constraining regional fluxes. The problem is addressed by identifying concentration spikes of few minutes duration in greenhouse gas continuous time series from 4 stations by applying automatic detection methods (COV, SD and REBS) previously used for atmospheric pollution but not systematically for greenhouse gas time series.

Language and structure

The article is well written, though too lengthy and it contains quite some minor but still sloppy errors that should be corrected. For this I included a list of minor corrections at the end of this review. An additional check of the text by a native speaker would be beneficial to the paper. Here and there the text is too long. It is a useful exercise, but not rocket science, so could be dealt with also by a shorter text. I propose to shorten section 3.3 and 3.4 with 30-50%. An important issue is that the paper only handles two methods as the COV method is discarded right away. The text should be revised to better reflect this. I would suggest to move the first paragraph of the conclusions in section 4 to replace parts of the introduction and summary, as this is the best introduction text to the paper.

**General comments**

The topic is very relevant for improving the quality of ambient greenhouse gas observations by a regional network like the ICOS atmosphere network in Europe by an automated procedure, additionally to human manual quality control. The methodology used is sound but not spectacular. The two spike detection methods tested are very basic and relatively straightforward techniques that have proven their usefulness in air quality applications. It would have been useful to also look into more sophisticated methods that apply Fourier transform Savitkzy-Golay (1964) filters or wavelet transforms (e.g. Wee et al, 2008) to achieve this end. I would like to see some good arguments whether and why this has not been considered. I agree with referee #2 that it would be good to refer to the percentages of hours detected than the absolute number. It would be good to state in the text more clear that avoiding spikes is more important than filtering them out and detection of spikes should always be followed by looking to the cause of the spikes in order to try to minimize them further. It is good to see from this paper that the contribution of the spikes in general is low on the average signal observed, except for the PDM site with the obvious problem of the nearby pollution source. The 4 sites chosen for the paper are said to be representative for the ICOS atmosphere station network, but neither of them is a continental tall tower

within 100 km or an urban region. It would also be interesting to see how the spike detection results vary for the vertical gradient along a tall tower where the footprint of the measurements varies from local for low sampling heights to more regional for the higher elevations.

Minor corrections

I24 European -> European Research

I28 in Amsterdam Island -> on Amsterdam Island

I38 change to: analyzers located at 200m from each other,

I40 we -> we also

I42 as -> in; for -> used for

143 like ICOS -> like that of the ICOS atmosphere network

I53 thereafter -> hereafter

I54 allows -> allows for

I54 move "to separate" after "time series"

I58 CO2 -> CO2,

I59 while -> because

l60 logbook -> a logbook

166 are -> is

l69 modelers -> modelers,

I77 been rarely -> rarely been

180-185 these sentences should be move to forward in introduction

**C3**

185 emissions -> emissions, instrumental failures, intermittent leaks etc.

190 European -> ICOS RI?

1101 are -> have been

1102 (Bergamaschi -> (e.g. Bergamaschi

1140 recommendations -> specification (https://www.icosri.eu/documents/ATC%20Public)

I161 calculations -> calculation

I212 and proved robustness -> and has been proven robust

I218 As all data in our study in the first step is averaged to 1 minute values

I247 remove ", spikes in other words"

l258 lead -> leads

I287-289 repetitive text

I291 detect automatically -> automatically detect

I291-293 As COV method is discarded move this to introduction and forget about it in the whole paper

I321 that -> that the

1338 remove "even"

1344 remove "the"

I368 methods -> methods that

I407 remove "The"

References

Savitzky, A.; Golay, M.J.E. (1964). Analytical Chemistry. doi:10.1021/ac60214a047 Wee et al (2008). Electrophoresis. doi:10.1002/elps.200800096

---

## Author Comment (AC1) · 6 Jan 2018

We would like to thank referee #2 for the valuable comments and her/his time to review this paper. Our answers to the points raised in this review are presented below.

Comment on section 3.1.2 1- Referee's comment: "In section 3.1.2 the authors indicate that test values ranging Beta from 1 to 10 had been done. The results of these tests should be presented (at least in the supplementary information). Otherwise, the affirmation that the optimal value is Beta=8 seems arbitrary."

2- Author's response: We agree with the referee that it is important to show the sensitivity of Beta values on the REBS method since the choice of Beta modify significantly the results. Our choice of Beta was mainly based on the comparisons between REBS method results and known spikes selected by site managers. For this study, we estimated that Beta=8 respond better to our need, even though we wish we could have more events clearly identified by the site managers to validate this choice.

3- Author's changes in manuscript: Line 273: We further compared these two values of Beta at the four stations every week for the year 2015 (from January to December). Line 278: Spike detection statistics for Beta ranging between 1 and 10 are presented in Table S1, and additional illustrations for Beta =1, 4, 8, and 10 are in figure S3. Table S1 and figure S3 are now available in the supplement.

Comment on section 3.2 1- Referee's comment: "In table 4 it could be interesting to show also the number of spikes manually reported by site managers, at least for the sites where this information is available. (in AMS are close to 0.1% for all species and 1% for CO2 according to text)."

2- Author's response: The main issue we are facing with the manual detection of spikes is the lack of systematic information. At most sites, there is no permanent staff, and consequently, the report of spikes with identified local processes are very sparse. We have used the few available periods, presented in section 3.4 for the Pic De Midi case (three months of winter 2015) and Finokalia (few weeks during the two years of the study) as case studies, but the comparison with the two years of systematic identification of spikes (Table 4) may be misleading because of the lack of completeness of the reports of the station managers.

3- Author's changes in manuscript: Line 286: The statistics for local spikes detection with the three methods are given in Table 4. Due to the lack of completeness of the reports by the staff about potential local contaminations, we cannot compare those average statistics to the manual spike detection.

Comment on section 3.4 1- Referee's comment: "The overestimation is not reflected in

the paper (not even in section 3.3). Probably is very low for both SD and REB methods (as in fact they underestimate the number of events) but needs to be quantified. For example, in section 3.4: was there any event not considered by manual spike detection that was considered as spike by any of the 2 detection methods?"

2- Author's response: It is very difficult to quantify possible overestimation of the number of spikes determined by the automatic algorithms, due to the non-systematic determination of spikes by the site managers, as explained in the previous point. We have run few additional tests to quantify the number of events not considered by the manual spike detection and considered by the automatic methods. For the Pic du Midi (section 3.4) the SD and the REBS methods detect a total of 3402 and 2981 contaminated data respectively with 211 (for SD) and 133 (for REBS) data not considered by manual detection. These events represent a percentage of 0.25% data considered by the SD method and not considered by the manual detection from December to March, and 0.15% for REBS method. It should be noted, however, that the manual spike detection information is not exhaustive in a sense that the person in charge does not necessarily have information on all contaminated events. Due to these missed detections, we think we cannot qualify these differences as overestimations.

3- Author's changes in manuscript: Line 397: We have also calculated the number of events not considered by the manual flagging and considered by the automatic methods. For a total of 3402 data detected by the SD method, only 211 data were not considered by the PI, which represents 0.25% on the whole period. For the REBS method, 133 data out of 2981 were not detected by the PI (nearly 0.15%). However, these statements should be used with caution since the manual spike detection information is not exhaustive, and the person in charge does not necessarily have information on all contaminated events.

Comment on section 3.4 1- Referee's comment: "In Table 6, it will be interesting to add the percentage for each classification on the whole series (e.g. OPE filtered data is, in 53% of cases, higher than 1ppm from non filtered. But which percentage of hours

this represents on the whole series?)." "In figure S6: please, use percentage of hours instead of number of hours. Graph will also be clearer with another X scale. Moreover, as in the text is said the figure is similar for other stations, it will be interesting to add the histograms of all sites."

2- Author's response: Accordingly, to the comment we have changed in table 6 the percentage on the impacted hours to the percentage on the whole time series. We agree that it is interesting to show the percentage of hours instead of the number. X scale were readjusted by fixing the number of intervals to 15. We assume this distribution will the graphs clearer.

3-Author's changes in manuscript: Table 6 is updated. Figure S6 is updated to figure S7, and it is now completed by results for all four sites.

Comment on bibliography 1- Referee's comment: The reference Foster et al. does not seems to be related with this article.

2- Author's response: We have corrected the reference list

---

## Author Comment (AC2) · 6 Jan 2018

We would like to thank referee#1 for the valuable comments and her/his time to review this paper. Author's comments are presented below.

1- Referee's comment: Language and structure The article is well written, though too lengthy and it contains quite some minor but still sloppy errors that should be corrected. For this I included a list of minor corrections at the end of this review. An additional check of the text by a native speaker would be beneficial to the paper. 1- Author's response: We thank the referee for pointing out the list of minor errors. We have corrected all the suggested errors.

[Figure]

2- Referee's comment: Here and there the text is too long. It is a useful exercise, but not rocket science, so could be dealt with also by a shorter text. I propose to shorten section 3.3 and 3.4 with 30-50%. 2- Author's response: As suggested by the referee we have shortened the sections 3.3 and 3.4.

3- Referee's comment: An important issue is that the paper only handles two methods as the COV method is discarded right away. The text should be revised to better reflect this. 3- Author's response: We agree with the referee that the COV method was not developed as much as the other two (SD and REBS) in the paper. The COV method showed its robustness to filter out spikes, yet it requires setting the spike percentage a priori. In the abstract (line 32) we explained that the COV method will be excluded from the case study analysis"Analysis of the results of each method leads us to exclude the COV method due to the requirement to arbitrarily specify an a-priori percentage of rejected data in the time series, which may over- or under-estimate the actual number of spikes.". In line 189 we said that this method is not automatic, and in figure S2 in the supplement, we presented that we could select the same spike as the SD and the REBS method if we set the right a-priori percentage used as an input for the COV method. In line 302 "Because of this limitation for automatic spike detection we have discarded the COV method from further tests for the selection of the most reliable method for spike detection." we explained why we discarded the method for the Pic du Midi and Finokalia case study comparisons. 3-Author's changes in manuscript: We have added this sentence to the manuscript. Line 97: The study will focus more on the SD and the REBS method since they are fully-automatic and they do not require any a-priori information for the implementation.

4- Referee's comment: I would suggest to move the first paragraph of the conclusions in section 4 to replace parts of the introduction and summary, as this is the best introduction text to the paper. 4- Author's response: We agree that the first paragraph of conclusion could be used in the introduction. 4-Author's changes in manuscript: We have moved the first paragraph of the conclusion in introduction line 75. We have

added few lines in the conclusion: line 429 "The recent increase in the number of studies that have been applied to study the spatial representativeness of GHG observations demonstrate the need to define efficient and reliable methods for the identification spikes related to local contamination sources. Three methods based on the standard deviation calculation were compared in order to provide an objective algorithm for the GHG data spike detection."

5- Referee's comment: General comments The topic is very relevant for improving the quality of ambient greenhouse gas observations by a regional network like the ICOS atmosphere network in Europe by an automated procedure, additionally to human manual quality control. The methodology used is sound but not spectacular. The two spike detection methods tested are very basic and relatively straightforward techniques that have proven their usefulness in air quality applications. It would have been useful to also look into more sophisticated methods that apply Fourier transform Savitkzy-Golay (1964) filters or wavelet transforms (e.g. Wee et al, 2008) to achieve this end. I would like to see some good arguments whether and why this has not been considered. 5- Author's response: We thank referee#1 for reflecting this issue. We looked into different methodology before selecting these methods. Considering the wavelet transforms (Wee et al, 2008) and the Fourier transform methods, both methods showed their robustness for filtering out spikes in earlier studies. However, the two methods require the same conditions: the signal should be continuous, and smooth. In our case none of these conditions are full filled. Considering that the measurements are regularly interrupted due to different reasons (e.g. calibration, flushing time after switching from sampling level to another, power or internet outage), we had to select a method that handles time-series with data gaps. Moreover, if we apply a Fourier Transform method on continuous measurements, we will get a signal composed with different frequencies only. All information that varies with the time will be lost. In other words, we can analyze what happen (spikes to be filtered out), without knowing when this happens. In fact, we consider that having also the time information related to the spike would allow us to better understand the origin of the contamination (as presented in the manuscript

for the Pic du Midi and Finokalia cases).

6- Referee's comment: I agree with referee #1 that it would be good to refer to the percentages of hours detected than the absolute number. 6- Author's response: We have changed in table 6 the percentage of the impacted hours to the percentage of the whole time series. We agree that it is interesting to show the percentage of hours instead of the number. Figure S7 is now completed by results for all sites. 6-Author's changes in manuscript: Table 6 is updated. Figure S6 is updated to figure S7.

7- Referee's comment: It would be good to state in the text more clear that avoiding spikes is more important than filtering them out and detection of spikes should always be followed by looking to the cause of the spikes in order to try to minimize them further. 7- Author's response: We agree that it is a major issue, and we have added in the conclusion (lines from 456 to 461) a sentence to emphasis this point. This problem was also reflected in the Pic du Midi case, where the use of the spike detection method confirms the need to move the inlet by 200m from its original location, in order to avoid the frequent contaminations. The aim of this study is to evaluate the methods in order to select the one that will be implemented in the ICOS Atmospheric Thematic Center Quality Control (ATC-QC) software to perform daily spike detection. In a future study, the chosen method will be coupled with meteorological data, such as wind direction, in order to try to assess the origin of the contaminations. 7-Author's changes in manuscript: (lines from 456 to 461): However, even if the implementation of an automatic algorithm can successfully identify short-term spikes due to local contaminations, it is important to note that the priority in the selection of a background site should be to avoid as much as possible the occurrence of such spikes. In the case where the spikes can not be totally avoided, it is then important to try to understand their cause and look for possible actions to minimize them. The modification of the air intake at the Pic du Midi, described in this study, is a very good example of what can be done once the origin of spikes is understood.

8- Referee's comment: It is good to see from this paper that the contribution of the

spikes, in general, is low on the average signal observed, except for the PDM site with the obvious problem of the nearby pollution source. The 4 sites chosen for the paper are said to be representative for the ICOS atmosphere station network, but neither of them is a continental tall tower within 100 km or an urban region. It would also be interesting to see how the spike detection results vary for the vertical gradient along a tall tower where the footprint of the measurements varies from local for low sampling heights to more regional for the higher elevations. 8- Author's response: The OPE station presented in this study is a 120m tall tower site located in a rural area, which is representative of most of the future ICOS tall towers. The measurements are carried out at three sampling heights 10, 50, and 120m. Every hour the ambient air is sampled for 20 min alternatively at the three levels. We have run the SD method separately on the three levels. The method detects for CH4 a percentage of 1.6%, 1.9% and 1.8% of contaminated data, and 1%, 1% and 1.1% for CO2 respectively for 10m, 50m, and 120m. The comparison between the three sampling height did not show a significant gradient along the tall tower. Moreover, the selected spikes did not occur during the same hours for the three levels. This is mainly related to the difference in the sampling time along the tall tower.